# Globally Interpretable Graph Learning via Distribution Matching

## ABSTRACT

Graphs neural networks (GNNs) have emerged as a powerful graph learning model due to their superior capacity in capturing critical graph patterns. Instead of treating GNNs as black boxes in an end-to-end fashion of training and deployment, people start to turn their attention to understand and explain the model behavior. Existing works mainly focus on local interpretation, which aims to reveal the discriminative pattern for individual instances. However, the retrieved pattern cannot be directly generalized to reflect the high-level model behavior, i.e., patterns captured by the model for a certain class. To gain global insights about graph learning mechanism, we aim to answer an important question that is not yet well studied: *how to provide a global interpretation for the graph learning procedure?* We formulate this problem as *globally interpretable graph learning*, which targets on distilling high-level and human-intelligible patterns that dominate the learning procedure, such that training on this pattern can recover a similar model.

To address this problem, we first propose a new interpretation metric *model fidelity*, which is tailored for evaluating the fidelity of the resulting model trained on interpretations. Our preliminary analysis shows that interpretative patterns generated by existing global methods fail to recover the model training procedure. Thus, we further propose our solution, *Graph Distribution Matching* (GDM), which synthesizes interpretive graphs by matching the distribution of the original and interpretive graphs in the feature space of the GNN as its training proceeds. These few interpretive graphs demonstrate the most informative patterns the model captures during training. Extensive experiments on graph classification datasets demonstrate multiple advantages of the proposed method, including high model fidelity, predictive accuracy and time efficiency, as well as the ability to reveal class-relevant structure.

## CCS CONCEPTS

• **Computing methodologies** → *Semantic networks*; • **Theory of computation** → *Semi-supervised learning*.

## KEYWORDS

Graph Neural Networks, Model Interpretability

**ACM Reference Format:**
Anonymous Author(s). 2018. Globally Interpretable Graph Learning via Distribution Matching. In *Proceedings of Make sure to enter the correct conference title from your rights confirmation emai (Conference acronym 'XX)*. ACM, New York, NY, USA, 11 pages. https://doi.org/XXXXXXX.XXXXXXX

## 1 INTRODUCTION

Graph neural networks (GNNs)[6, 11, 12, 23, 26, 27] have attracted enormous attention and prominently advanced the state of the art on graph learning tasks. Despite their great success, GNNs are usually treated as black boxes in an end-to-end fashion of training and deployment, which may raise trustworthiness concerns in decision making, if humans cannot really understand what pattern are really captured by the model during graph learning procedure. Lack of such understanding could be particularly risky when using a GNN model for high-stakes domains, e.g., finance [29] and medicine [5]. For instance, in the context of predicting the effect of medicines, if a GNN model mistakenly learns false patterns that violate chemical principles, it may provide incorrect assessments. This highlights the importance of ensuring a comprehensive interpretation of the working mechanism for graph learning.

To improve transparency of GNNs, a large body of existing interpretation techniques focuses on providing *instance-level local interpretation*, which explains specific predictions a GNN model makes on each individual graph instance [1, 3, 8, 15, 18, 19, 21, 24, 28, 32]. Despite different strategies adopted in these works, in general, local interpretation aims to identify critical substructure for a particular graph instance, which would require manual inspections on many local interpretations to mitigate the variance across instances and conclude a high-level pattern of the model behavior. As a sharp comparison to such instance-specific interpretations, relatively few recent works study *model-level global interpretations* [2, 25, 30] to understand the general behavior of the model with respect to a certain class instead of any particular instance.

The goal of global interpretation is to generate a few compact interpretive graphs, which summarize class discriminative patterns the GNN model learns for decision making. Existing works generate such interpretive graphs via different strategies, including reinforcement learning [30], concept combination [2] and probabilistic generation [25]. These solutions can extract meaningful interpretive graphs with a high *predictive accuracy*, evaluated from the perspective of *model consumers*: given a pre-trained GNN model, the end user can use these interpretation methods to understand what patterns this model is leveraging for *inference*.

In this paper, we aim to interpret at the side of *model developers/providers*, who usually care about what patterns really dominate the model training, which could help improve training transparency. This demands specialized evaluation, which are long ignored: if the interpretation indeed contains essential patterns the model captures during training, then when we use these interpretive graphs to train a model from scratch, this surrogate model should present similar behavior as the original model. We are the first to realize this principle and define a new metric, *model fidelity*, which evaluates the predictive similarity between the surrogate model (trained via interpretative graphs) and the original model (normally trained via the training set). We evaluate model fidelity of existing global interpretation method, XGNN [30] and GNNInterpreter [25], by comparing the surrogate model and the original model for each


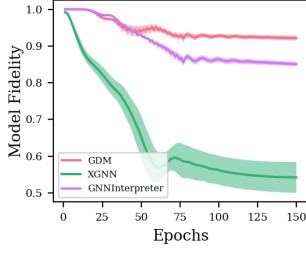
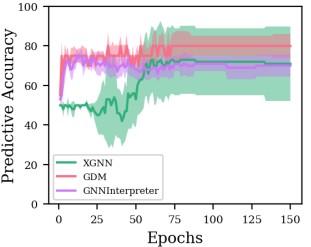

**Figure 1:** *Model Fidelity* **(i.e., cosine similarity between the predictive logits of original model and that of surrogate model) and** *Predictive Accuracy* **(i.e., the original model's accuracy on interpretive graphs) as model training proceeds.**

training iteration on MUTAG data. As shown in Figure 1, they have relatively low model fidelity, which suggests that their generated interpretive graphs are less discriminative for recovering the original model's training procedure. Thus these interpretation methods may not suit for explaining model training behavior.

To this end, we attempt to provide a novel globally interpretable graph learning framework, which is designed for the model developers to distill high-level and human-intelligible patterns the model learns in its training procedure. To be more specific, we propose Graph Distribution Matching (GDM) to synthesize a few compact interpretive graphs for each class following the *distribution matching principle*: as the model training progresses, the interpretive graphs maintain a similar distribution as the original graphs. We optimize interpretive graphs by minimizing the distance between the interpretive and original data distributions, measured as the maximum mean discrepancy (MMD) [7] in a family of embedding spaces obtained by a series of model snapshots. Presumably, GDM simulates the model training trajectory, thus the generated interpretation can provide a general understanding of what patterns dominate and result in the model training behavior.

Note that as model developer, we can access the model training trajectory, and our proposed framework is an efficient plug-and-play interpretation tool that can be easily integrated to usual model development pipeline, without interfering the normal training procedure. The success of our framework enables the model develops to provide an interpretation byproduct when publishing their models, which can benefits multiple parties: for the developers, models are published with better transparency without leaking training data; for the consumers, the interpretation can help screen whether the models' discriminative patterns fit their needs.

Extensive quantitative evaluations on three synthetic and three real-world datasets for graph classification task verify the effectiveness of GDM: it can simultaneously achieve high model fidelity and predictive accuracy. Our ablation study also shows the advantage of generating interpretation guided by the model training trajectory. Qualitative study further intuitively demonstrates the human-intelligible patterns captured by GDM.

## 2 RELATED WORK

Due to the prevalence of graph neural networks (GNNs), extensive efforts have been conducted to improve their transparency and

interpretability. Existing techniques can be categorized as *local instance-level* interpretation and *global model-level* interpretation depending on the interpretation form.

### 2.1 Local Instance-Level Interpretation

Instance-level methods provide input-dependent explanations for each individual graph [1, 32]. Given an input graph, these methods explain GNNs by extracting a small interpretive subgraph. Existing solutions can be categorized as gradient-based [3, 19], attention-based [18], perturbation-based [16, 28], decomposition-based [8], and surrogate-based methods [24]. Gradient-based method directly uses the gradients as the approximations of feature importance. Attention-based methods use the attention mechanism to identify important subgraph as interpretation. Perturbation-based methods optimize a subgraph mask to captures the important nodes and edges. Surrogate-based explanation methods use data sampling to filter out unimportant features and an explainable small model — such as a probabilistic graphical model — is fitted on the filtered data as a topological explanation. Decomposition-based methods decompose predictive scores to represent how importance the input contributes to the predicted results. Again, instance-level methods are based on each input instance. Although they are helpful for getting an explanation for every single graph, they can hardly capture the commonly important features that are shared by graph instances for each class. Therefore, it is necessary to have both instance-level and model-level interpretations for GNNs.

### 2.2 Global Model-Level Interpretations

Model-level interpretation aims at capturing the global behaviour of the model as a whole, such that a robust overview of the model can be summarized from individual noisy local explanations. This type of interpretation on graph learning is less studied. XGNN [30] frames this problem as a form of input optimization, leveraging a reinforcement learning technique to sequentially generate edges based on the prediction reward. However, this approach requires domain expert knowledge to design valid reward function for different inputs, which is not always available. GNNInterpreter [25] learns a probabilistic generative graph distribution and identifies the key graph pattern when GNN tries to make a certain prediction. GLGExplainer [2] generates explanations as Boolean combinations of learned graphical concepts, represented as clusters of local explanations. While these methods identify intuitive class-related patterns that can be recognized by the model (with high predictive accuracy), they usually ignores the training utility of these explanations. Ideally, high-quality interpretations capturing class discriminative patterns from the training data should be able to train a similar model. From this perspective, in this work, we define model fidelity as a new metric, and propose a novel globally interpretable graph learning framework that explains by matching the distribution along the model training trajectory.

## 3 METHODS

We first discuss existing global training methods and provide a general form of the targeted problem. To improve the utility of class discriminative explanations in training a similar model, we propose a novel globally interpretable graph learning framework.

This framework aims to align the model's behavior on original training data and synthesized interpretive data along the model training trajectory. We realize this goal via the distribution matching principle, which can be formulated as an optimization problem. We further discuss several practical constraints for optimizing interpretive graphs. Finally, we provide the designed algorithm for the proposed interpretation method.

### 3.1 Graph Learning Background

We focus on explaining GNNs' global behavior for the graph classification task. A graph classification dataset with $N$ graphs can be denoted as $\mathcal{G} = \{G^{(1)}, G^{(2)}, \dots, G^{(N)}\}$ with a corresponding ground-truth label set $\mathcal{Y} = \{y^{(1)}, y^{(2)}, \dots, y^{(N)}\}$. Each graph consists of two components, $G^{(i)} = (\mathbf{A}^{(i)}, \mathbf{X}^{(i)})$, where $\mathbf{A}^{(i)} \in \{0, 1\}^{n \times n}$ denotes the adjacency matrix and $\mathbf{X}^{(i)} \in \mathbb{R}^{n \times d}$ is the node feature matrix. The label for each graph is chosen from a set of $C$ classes $y^{(i)} \in \{1, \dots, C\}$, and $y_c^{(i)}$ denotes that the label of graph $G_i$ is $c$, that is $y^{(i)} = c$. A set of graphs that belong to class $c$ could be further represented as $\mathcal{G}_c = \{G^{(i)} | y^{(i)} = c\}$.

A GNN model $\Phi(\cdot)$ is a concatenation of a feature extractor $f_\theta(\cdot)$ parameterized by $\theta$ and a classifier $h_\psi(\cdot)$ parameterized by $\psi$, where $\Phi(\cdot) = h_\psi(f_\theta(\cdot))$. The feature extractor $f_\theta : \mathcal{G} \rightarrow \mathbb{R}^{d'}$ takes in a graph and embeds it to a low-dimensional space with $d' \ll d$. The classifier $h_\psi : \mathbb{R}^{d'} \rightarrow \{1, \dots, C\}$ further outputs the predicted class given the graph embedding.

### 3.2 Revisit Global Interpretation Problem

We now provide a general form for the global interpretation problem. The idea is to generate a small set of compact graphs that can explain the high-level behavior of the GNN model, e.g., what patterns lead the model to discriminate different classes. Specifically, given a GNN model $\Phi^*$, exsiting global interpretation method aims to generate interpretive graphs that have the maximal predicted probability for a certain class $y_c$. Formally, this problem can be defined as:

$$\min_{\mathcal{S}} \mathcal{L}(\Phi^*(\mathcal{S}), y_c), \tag{1}$$

where $\mathcal{S}$ is one or multiple compact interpretive graphs capturing key graph structures and node characteristics for interpretation, and $\mathcal{L}(\cdot, \cdot)$ is the loss (e.g., cross-entropy loss) of predicting $\mathcal{S}$ as label $y_c$. Existing global interpretation techniques can fit in this form but differ in the generation procedure of $\mathcal{S}$. For instance, in XGNN [30], $\mathcal{S}$ is defined as a set of completely synthesized graphs with each edge generated by a reinforcement learning strategy. The goal of the reward function is to maximize the probability of predicting $\mathcal{S}$ as a certain class $y_c$. In GNNInterpreter [25], $\mathcal{S}$ is generated by sampling from an estimated graph distribution. In GLGExplainer [2], $\mathcal{S}$ is generated by a Boolean logic function. Despite their difference in generation techniques, they stand on a common ground as a model consumer: they can only access and inspect the final pre-trained model $\Phi^*$ to explain its behavior.

If standing from the perspective of model provider, such a problem formulation may not fully leverage all accessible information, such as the the whole training trajectory, leading to limited interpretation capability. Specifically, we consider interpretation quality from the following two aspects:

- *Predictive Accuracy* reflects whether the extracted interpretative patterns are really class-relevant. It is calculated as the model accuracy on generated interpretive graphs. Existing works mainly focus on this aspect [2, 25, 30].
- *Model Fidelity* measures whether the interpretive graphs are class discriminative enough to train a similar model. It is calculated as the cosine similarity between the predictive probabilities of the target model and that of the surrogate model (trained by interpretative graphs) on a same set of instances. This aspect however has never been inspected in prior studies.

As shown in Figure 1, existing works following this formulation provide a limited model fidelity. This observation motivates us to rethink the global interpretation problem from the model provider's perspective and design a globally interpretable learning framework.

### 3.3 Globally Interpretable Graph Learning

Our goal is to generate global explanations that can not only be accurately predicted as the corresponding class, but also lead to a high-fidelity model. In order to achieve this goal, we propose to optimize the explanations in the model developing stage, such that the training trajectory information can be leveraged. We thus propose a novel research problem: *how to provide global interpretation for a model training procedure, such that training on such interpretation can recover a similar model?* We frame this problem as *globally interpretable graph learning*, which can be defined as the following optimization problem:

$$\min_{\mathcal{S}} \mathbb{E}_{t \sim \mathcal{T}} [\mathcal{L}(\Phi_t(\mathcal{S}), y_c)],$$
$$\text{s.t. } \Phi_t = \text{opt} - \text{alg}_\Phi(\mathcal{L}_{\text{CE}}(\Phi_{t-1}), \varsigma), \tag{2}$$

where $\mathcal{T} = [0, \dots, T-1]$ is the normal training iterations for the target GNN model, and $\text{opt} - \text{alg}$ is a specific model update algorithm (e.g., gradient descent) with a fixed number of steps $\varsigma$. $\mathcal{L}_{\text{CE}}(\Phi) = \mathbb{E}_{G, y \sim \mathcal{G}, \mathcal{Y}} [\ell(\Phi(G), y)]$ is the cross-entropy loss used for normal GNN model training.

This formulation of globally interpretable graph learning states that the interpretable patterns $\mathcal{S}$ should be optimized based on the whole training trajectory $\Phi_0 \rightarrow \Phi_1 \rightarrow \dots \rightarrow \Phi_{T-1}$ of the model. This stands in sharp contrast to other global interpretation where only the final model $\Phi^* = \Phi_{T-1}$ is considered. The training trajectory reveals more information about model's training behavior to form a constrained model space, such as essential graph patterns that dominate the training of this model.

### 3.4 Interpretation via Distribution Matching

To realize globalinterpretation as demonstrated in Eq. (2), we now introduce the exact form of the objective function for optimizing interpretive graphs that encapsulate the model's learning behavior from the data. Recall that a GNN model is a combination of feature extractor and a classifier. The feature extractor $f_\theta$ usually carries the most essential information about the model, while the classifier is a rather simple multi-perceptron layer. Since the feature extractor plays the majority role, a natural idea for generating interpretation is to match its distribution with training graphs in the model's feature space. We name this interpretation principle as *Graph Distribution Matching* (GDM).

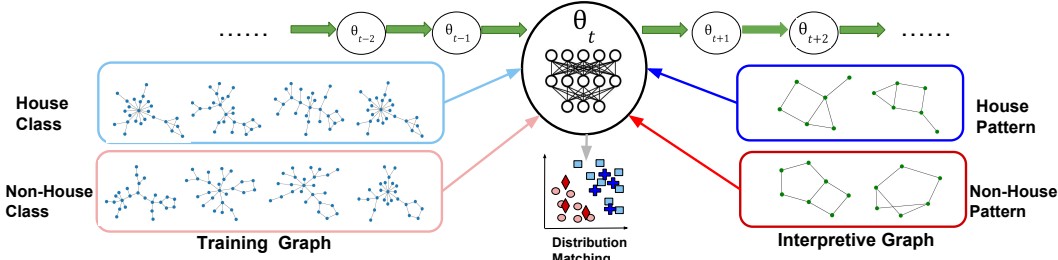

**Figure 2: Overview of the proposed globally interpretable learning framework via graph distribution matching GDM.**

**Graph Distribution Matching (GDM)** To realize this principle, we first measure the distance between two graph distributions via their maximum mean discrepancy (MMD), which is the difference between means of distributions in a Hilbert kernel space $\mathcal{H}$ [7]:

$$\sup_{\|f_{\boldsymbol{\theta}}\|_{\mathcal{H}} \leq 1} \left( \mathbb{E}_{G \sim \mathcal{G}_c} [f_{\boldsymbol{\theta}}(G)] - \mathbb{E}_{S \sim \mathcal{S}_c} [f_{\boldsymbol{\theta}}(S)] \right). \quad (3)$$

Empirically, MMD can be estimated as the difference between the encoded training graphs and interpretive graphs in the embedding space. Based on this idea, we instantiate the outer objective in Eq. (2) as a distribution matching loss $\mathcal{L}_{\text{DM}}(\cdot)$:

$$\mathcal{L}(\Phi_t(\mathcal{S}), y_c) \coloneqq \mathcal{L}_{\text{DM}}(f_{\boldsymbol{\theta}_t}(\mathcal{S}_c))$$

$$= \| \frac{1}{|\mathcal{G}_c|} \sum_{G \in \mathcal{G}_c} f_{\boldsymbol{\theta}_t}(G) - \frac{1}{|\mathcal{S}_c|} \sum_{S \in \mathcal{S}_c} f_{\boldsymbol{\theta}_t}(S) \|^2, \quad (4)$$

where $\mathcal{S}_c$ is the interpretive graph(s) for explaining class $c$, and $\mathcal{G}_c$ is the training graphs belonging to class $c$. By optimizing Eq. (4), we can obtain interpretive graphs that produce similar embeddings to training graphs for the current GNN feature extractor $\boldsymbol{\theta}_t$ in the training trajectory. Thus, the interpretive graphs provide a plausible explanation for the model learning process. Note that there can be multiple interpretive graphs for each class, i.e., $|\mathcal{S}_c| \geq 1$. With this approach, we are able to generate an arbitrary number of interpretive graphs that capture different patterns.

**Globally Interpretable Learning via Distribution Matching** By plugging the distribution matching objective Eq. (4) into Eq. (2), and simultaneously optimizing interpretive graphs for multiple classes $\mathcal{S} = \{\mathcal{S}_c\}_{c=1}^{C}$, we can rewrite our learning goal as follows:

$$\min_{\mathcal{S}} \mathbb{E}_{t \sim \mathcal{T}} \big[ \sum_{c=1}^{C} \mathcal{L}_{\text{DM}}(f_{\boldsymbol{\theta}_t}(\mathcal{S}_c)) \big]$$

$$\text{s.t. } \boldsymbol{\theta}_t, \boldsymbol{\psi}_t = \text{opt} - \text{alg}_{\boldsymbol{\theta}, \boldsymbol{\psi}}(\mathcal{L}_{\text{CE}}(h_{\boldsymbol{\psi}_{t-1}}, f_{\boldsymbol{\theta}_{t-1}}), \varsigma), \quad (5)$$

where the cross entropy loss is w.r.t. the feature extractor and predictive head, $\mathcal{L}_{\text{CE}}(\Phi) = \mathcal{L}_{\text{CE}}(h_{\boldsymbol{\psi}}, f_{\boldsymbol{\theta}}) = \mathbb{E}_{G, y \sim \mathcal{G}, \mathcal{Y}} [(h_{\boldsymbol{\psi}}(f_{\boldsymbol{\theta}}(G)), y)]$, and for each class $c$, we optimize its corresponding interpretive graph(s) $\mathcal{S}_c$. The interpretation procedure is based on the model training trajectory, while the model is normally trained on the original classification task. Thus this interpretation method can serve as a plug-and-play tool without interfering normal model training.

The proposed framework is illustrated in Figure 2, for each training step $t$, we update interpretive graphs by aligning with the training graphs in the GNN model's feature space via distribution matching. Along the whole training trajectory, we keep updating

interpretive graphs in a curriculum learning manner to capture the model's training behavior. It is worth noting that such a distribution matching scheme has shown success in distilling rich knowledge from training data to synthetic data [33], which preserve sufficient discriminative information for training the underlying model. This justifies our design of distribution matching for interpretation.

## 3.5 Practical Constraints in Graph Optimization

Optimizing each interpretive graph is essentially optimizing its adjacency matrix and node feature. Denote a interpretive graph as $S = (\mathbf{A}_s, \mathbf{X}_s)$, with $\mathbf{A}_s \in \{0, 1\}^{m \times m}$ and $\mathbf{X}_s \in \mathbb{R}^{m \times d}$. To generate solid graph explanations using Eq. (5), we introduce several practical constraints on $\mathbf{A}_s$ and $\mathbf{X}_s$. The constraints are applied on each interpretive graph, concerning discrete graph structure, matching edge sparsity, and feature distribution with the training data.

**Discrete Graph Structure** Optimizing the adjacency matrix is challenging as it has discrete values. To address this issue, we assume that each entry in matrix $\mathbf{A}_s$ follows a Bernoulli distribution $\mathcal{B}(\Omega)$ : $p(\mathbf{A}_s) = \mathbf{A}_s \odot \sigma(\Omega) + (1 - \mathbf{A}_s) \odot \sigma(-\Omega)$, where $\Omega \in [0, 1]^{m \times m}$ is the Bernoulli parameters, $\sigma(\cdot)$ is element-wise sigmoid function and $\odot$ is the element-wise product, following [9, 13, 14]. Therefore, the optimization on $\mathbf{A}_s$ involves optimizing $\Omega$ and then sampling from the Bernoulli distribution. However, the sampling operation is non-differentiable, thus we employ the reparameterization method [17] to refactor the discrete random variable into a function of a new variable $\varepsilon \sim \text{Uniform}(0, 1)$. The adjacency matrix can then be defined as a function of Bernoulli parameters as follows, which is differentiable w.r.t. $\Omega$:

$$\mathbf{A}_s(\Omega) = \sigma((\log \varepsilon - \log(1 - \varepsilon) + \Omega)/\tau), \quad (6)$$

where $\tau \in (0, \infty)$ is the temperature parameter that controls the strength of continuous relaxation: as $\tau \to 0$, the variable $\mathbf{A}_s$ approaches the Bernoulli distribution. Now Eq. (6) turns the problem of optimizing the discrete adjacency matrix $\mathbf{A}_s$ into optimizing the Bernoulli parameter matrix $\Omega$.

**Matching Edge Sparsity** Our interpretive graphs are initialized by randomly sampling subgraphs from training graphs, and their adjacency matrices will be freely optimized, which might result in too sparse or too dense graphs. To prevent such scenarios, we exert a sparsity matching loss by penalizing the distance of sparsity between the interpretive and the training graphs, following [9]:

$$\mathcal{L}_{\text{sparsity}}(\mathcal{S}) = \sum_{(\mathbf{A}_s(\Omega), \mathbf{X}_s) \sim \mathcal{S}} \max(\bar{\Omega} - \epsilon, 0), \quad (7)$$

**Algorithm 1** Globally Interpretable Learning via Graph Distribution Matching (GDM)

1: **Input**: Training data $\mathcal{G} = \{\mathcal{G}_c\}_{c=1}^C$
2: Initialize explanation graphs $\mathcal{S} = \{\mathcal{S}_c\}_{c=1}^C$ for each class $c$
3: **for** $t = 0, \ldots, T-1$ **do**
4:     Sample mini-batch interpretive graphs $B^{\mathcal{S}} = \{B_c^{\mathcal{S}} \sim \mathcal{S}_c\}_{c=1}^C$
5:     Sample mini-batch training graphs $B^{\mathcal{G}} = \{B_c^{\mathcal{G}} \sim \mathcal{G}_c\}_{c=1}^C$
6:     *# Optimize global interpretive graphs*
7:     **for** class $c = 1, \ldots, C$ **do**
8:         Compute the interpretation loss following Eq. (9): $\mathcal{L}_c = \mathcal{L}_{\text{DM}}(f_{\boldsymbol{\theta}_t}(B_c^{\mathcal{S}})) + \alpha \cdot \mathcal{L}_{\text{feat}}(B_c^{\mathcal{S}}) + \beta \cdot \mathcal{L}_{\text{sparsity}}(B_c^{\mathcal{S}})$
9:     **end for**
10:     Update explanation graphs $\mathcal{S} \leftarrow \mathcal{S} - \eta \nabla_{\mathcal{S}} \sum_{c=1}^C \mathcal{L}_c$
11:     *# Optimize GNN model as normal*
12:     Compute normal training loss for graph classification task $\mathcal{L}_{\text{CE}}(h_{\boldsymbol{\psi}_{t-1}}, f_{\boldsymbol{\theta}_{t-1}}) = \sum_{G \sim B^{\mathcal{G}}} (h_{\boldsymbol{\psi}_{t-1}}(f_{\boldsymbol{\theta}_{t-1}}(G)), y)$
13:     Update feature extractor $\boldsymbol{\theta}_{t+1} = \boldsymbol{\theta}_t - \eta_1 \nabla_{\boldsymbol{\theta}} \mathcal{L}_{\text{CE}}(h_{\boldsymbol{\psi}_{t-1}}, f_{\boldsymbol{\theta}_{t-1}})$
14:     Update predictive head $\boldsymbol{\psi}_{t+1} = \boldsymbol{\psi}_t - \eta_2 \nabla_{\boldsymbol{\psi}} \mathcal{L}_{\text{CE}}(h_{\boldsymbol{\psi}_{t-1}}, f_{\boldsymbol{\psi}_{t-1}})$
15: **end for**
16: **Output**: Explanation graphs $\mathcal{S}^* = \{\mathcal{S}_c^*\}_{c=1}^C$ for each class $c$

where $\bar{\Omega} = \sum_{ij} \sigma(\Omega_{ij})/|\Omega|$ calculates the expected sparsity of a interpretive graph, and $\epsilon$ is the average sparsity of initialized $\sigma(\Omega)$ for all interpretive graphs, which are sampled from original training graphs thus resembles the sparsity of training dataset.

**Matching Feature Distribution** Real graphs in practice may have skewed feature distribution; without constraining the feature distribution on interpretive graphs, rare features might be overshadowed by the dominating ones. For example, in the molecule dataset MU-TAG, node feature is the atom type, and certain node types such as Carbons dominate the whole graphs. Therefore, when optimizing the feature matrix of interpretive graphs for such unbalanced data, it is possible that only dominating node types are maintained. To alleviate this issue, we propose to match the feature distribution between the training graphs and the interpretive ones.

Specifically, for each graph $G = (\mathbf{A}, \mathbf{X})$ with $n$ nodes, we estimate the graph-level feature distribution as $\bar{\mathbf{x}} = \sum_{i=1}^n \mathbf{X}_i / n \in \mathbb{R}^d$, which is essentially a mean pool of the node features. For each class $c$, we then define the following feature matching loss:

$$\mathcal{L}_{\text{feat}}(\mathcal{S}_c) = \| \frac{1}{|\mathcal{G}_c|} \sum_{(\mathbf{A}, \mathbf{X}) \in \mathcal{G}_c} \bar{\mathbf{x}} - \frac{1}{|\mathcal{S}_c|} \sum_{(\mathbf{A}_s, \mathbf{X}_s) \in \mathcal{S}_c} \bar{\mathbf{x}}_s \|^2, \quad (8)$$

where we empirically measure the class-level feature distribution by calculating the average of graph-level features. By minimizing the feature distribution distance in Eq. (8), even rare features can have a chance to be distilled in the interpretive graphs.

## 3.6 Final Objective and Algorithm

Integrating the practical constraints discussed in Section 3.5 with the distribution matching based interpretation framework in Eq. (5), we now obtain the final objective for interpretation optimization, which essentially is determined by the Bernoulli parameters for sampling discrete adjacency matrices and the node feature matrices.

Formally, we aims to solve the following optimization problem:

$$\min_{\mathcal{S}} \mathbb{E}_{t \sim \mathcal{T}} \Big[ \sum_{c=1}^C \mathcal{L}_{\text{DM}}(f_{\boldsymbol{\theta}_t}(\mathcal{S}_c)) + \alpha \cdot \mathcal{L}_{\text{feat}}(\mathcal{S}_c) + \beta \cdot \mathcal{L}_{\text{sparsity}}(\mathcal{S}) \Big]$$

$$\text{s.t. } \boldsymbol{\theta}_t, \boldsymbol{\psi}_t = \text{opt} - \text{alg}_{\boldsymbol{\theta}, \boldsymbol{\psi}}(\mathcal{L}_{\text{CE}}(h_{\boldsymbol{\psi}_{t-1}}, f_{\boldsymbol{\theta}_{t-1}}), \varsigma) \quad (9)$$

where we use $\alpha$ and $\beta$ to control the strength of regularizations on feature distribution matching and edge sparsity respectively. Algorithm 1 details the steps for solving this optimization problem.

**Complexity Analysis** We now analyze the time complexity of the proposed method. Suppose for each iteration, we sample $B_1$ interpretive graphs and $B_2$ training graphs. Denote their average edge number as $m$. The inner loop for interpretive graph update takes $m(B_1 + B_2)$ computations on node, while the update of GNN model uses $mB_2$ computations. Therefore the overall complexity is $O(mT(B_1 + 2B_2))$, which is of the same magnitude of complexity for normal GNN training. This demonstrates the efficiency of our interpretation method: it can simultaneously generate interpretations as the training of GNN model proceeds, without introducing extra complexity.

## 4 EXPERIMENTAL STUDIES

This section aims to verify the necessity of our proposed method for globally interpretable graph learning. Specifically, we conduct extensive experiments to answer the following questions:

- **Q1**: Does the proposed global interpretation result in similar GNN models as trained in original data (i.e., with high fidelity)?
- **Q2**: Is the training trajectory necessary for accurate global interpretation (compared with ensemble model snapshots)?
- **Q3**: Are the generated interpretations human-intelligible?

We provide both quantitative and qualitative study to evaluate the global interpretations generated by GDM, comparing with existing global interpretation baselines and ablation variants.

## 4.1 Experimental Setup

**Dataset** The interpretation performance is evaluated on the following synthetic and real-world datasets for graph classification, whose statistics can be found in Table 1.

- *Real-world* data includes: **MUTAG** [4] consists of chemical compounds with atoms as nodes and chemical bonds as edges, labeled by whether it has a mutagenic effect on a bacterium. **Graph-Twitter** [22] includes Twitter comments for sentiment classification with three classes. Each comment sequence is presented as a graph, with word embedding as node feature. **Graph-SST5** [31] is a similar dataset with reviews, where each review is converted to a graph labeled by one of five rating classes.
- *Synthetic* data includes: **Shape** contains four classes, i.e., Lollipop, Wheel, Grid, and Star. Each class has the same number of synthesized graphs with a random number of nodes. **BA-Motif** [16] uses Barabasi-Albert (BA) graph as base graphs, among which half graphs are attached with a "house" motif and the rest with "non-house" motifs. **BA-LRP** [20] based on Barabasi-Albert (BA) graph includes one class being node-degree concentrated graphs, and the other degree-evenly graphs. These datasets do not have node features, thus we use node index as the surrogate feature.

**Table 1: Basic Graph Statistics**

| Dataset | #Graph | #Node | #Edge | #Class | GCN Accuracy |
|---------|--------|-------|-------|--------|--------------|
| BA-Motif | 1000 | 25 | 50.93 | 2 | 100.00 |
| BA-LRP | 20000 | 20 | 42.04 | 2 | 97.95 |
| Shape | 100 | 53.39 | 813.93 | 4 | 100.00 |
| MUTAG | 188 | 17.93 | 19.79 | 2 | 88.63 |
| Graph-Twitter | 4998 | 21.10 | 40.28 | 3 | 61.40 |
| Graph-SST5 | 8544 | 19.85 | 37.78 | 5 | 44.39 |

**Baseline** We mainly compare GDM with global interpretation baselines, and ablative variants of our method.

- *Global interpretation baselines*: **XGNN** [30] generate global interpretation via reinforcement learning. Since it heavily relies on domain knowledge (e.g. chemical rules) in the reward function, thus we only evaluate it on MUTAG. **GNNInterpreter** [25] generates interpretations based on label and embedding similarity but it is only based on a pre-trained GNN model[1]. We also include a simple **Random** strategy as a reference, which randomly selects graphs from the training set as interpretations.
- *Ablation variants of GDM:* We also consider the variants of GDM which generate interpretation based on selective model snapshots. **GDM-First** and **GDM-Last** uses only the first or the last model snapshot respectively for the outer optimization in Eq. (5). **GDM-Ensemble** uses the same set of model snapshots as in GDM for conducting the outer optimization of Eq. (5), but ignores the sequence of model trajectory (i.e., disabling the inner optimization).

Meanwhile, a comparison of GDM with several local interpretation methods (which extract interpretive graphs for each training instance) can be found in Appendix A.1. A simple inherently global-interpretable method is also compared in Appendix A.2.

**Evaluation Protocol** We comprehend global interpretability from two perspectives, i.e., the interpretation should lead to high-fidelity model that is similar to the original target model (i.e., the model to be explained), and should have high chance to be predicted as the right classes. Based on this intuition, we establish the following evaluation protocols accordingly:

- *Model Fidelity* aims to verify whether the generated interpretation indeed captures essential class-discriminative patterns, such that the interpretation can be utilized to train a similar model as if it is trained on the original training set. Desired interpretation should capture patterns that dominate the model training procedure. To calculate this metric, we first use the generated interpretive graphs to train a surrogate model (with the same architecture as the original model) from scratch. Then we calculate model fidelity as the ratio of cases when the surrogate model makes the same decision as the orginal model on test data.
- *Model Utility* is to investigate whether the interpretation can lead to a high-utility model. Similarly, we train a surrogate model on the interpretation graphs. Then model utility is calculated as the surrogate model's predictive accuracy on test data.

---

[1]Since the official codebase is not yet available, its evaluation is based on our implementation following the paper

- *Predictive Accuracy* is to validate whether the interpretation can be correctly perceived by the target model as its corresponding class. Ideal interpretive graphs should be correctly classified to their classes by the target model being explained. We report the target model's predictive accuracy on the interpretive graphs as predictive Accuracy.

**Configurations** We choose the graph convolution network (GCN) as the target GNN model for interpretation, as it is widely used for graph learning. It contain 3 layers with 256 hidden dimension, concatenated by a mean pooling layer and a dense layers in the end. Adam optimizer [10] is adopted for model training. In both evaluation protocols, we split the dataset as 85% training and 15% test data, and only use the training set to generate interpretative graphs. To learn interpretive graphs that generalize to a distribution of model initializations, we empirically adopt regular model restarts to sample multiple trajectories. Given the interpretative graphs, each evaluation experiments are run 5 times, with the mean and variance reported.

## 4.2 Quantitative Results

This evaluation aims to answer the first question **Q1**. Meanwhile, we also report the commonly adopted predictive accuracy.

**Model Fidelity and Model Utility Performance** In Table 2, we compare GDM with baselines in terms of model fidelity and utility. XGNN performed on MUTAG achieves 89.47 fidelity and 68.40 utility with 10 graphs per class. We observe that GDM achieves remarkably better performance almost on all datasets, which indicates that GDM indeed captures discriminative patterns the model learns during training, such that our generated interpretation can also train a similarly useful model (with high model fidelity and utility). Meanwhile, different from XGNN, we do not include any dataset specific rules, thus is a more general interpretation solution.

**Predictive Accuracy** In Table 3, we compare the predictive accuracy of GDM, XGNN and GNNInterpreter respectively. Note that the predictive accuracy for GDM on all datasets except MUTAG is largen than 90%, implying that the generated graphs could preserve those essential information of the data, which plays a crucial role in guiding the desicion-making. Comparatively, GNNInterpreter has worse performance on most datasets, including Graph-Twitter, Graph-SST5, BA-LRP, and MUTAG, which indicates that several significant patterns of the data during training trajectory are lost and GNNInterpreter could not recover those undisclosed information along the training trajectory.

**Efficiency** Another advantage of GDM is that it generates interpretations in an efficient manner. As shown in Appendix A.3, GDM is almost 4 times faster than the global interpretation method XGNN. Our methods takes almost no extra cost to generate multiple interpretative graphs, as there are only few interpretive graphs compared with the training dataset. XGNN, however, select each edge in each graph by a reinforcement learning policy which makes the interpretation process rather expensive.

## 4.3 Model Analysis

**Ablation Study** In Table 4, we generate 10 interpretive graphs per class based on model snapshots. Intuitively, only using the first

**Table 2: *Model Fidelity* and *Model Utility* on a varying number of interpretive graphs generated per class.**

| Dataset | Graphs/Cls | Model Fidelity | | | Model Utility | | | GCN Accuracy |
|---|---|---|---|---|---|---|---|---|
| | | GDM | GNNInterpreter | Random | GDM | GNNInterpreter | Random | |
| MUTAG | 1 | **81.05 ± 9.76** | 79.53 ± 2.58 | 49.47 ± 10.84 | **71.92 ± 2.48** | 70.17 ± 2.48 | 50.87± 15.0 | |
| | 5 | **92.63 ± 2.58** | 84.21 ± 0.00 | 65.26 ± 6.31 | 77.19 ± 4.96 | 57.89 ± 4.29 | **80.70 ± 2.40** | 88.63 |
| | 10 | **94.73 ± 0.00** | 85.26 ± 6.14 | 66.31±5.37 | **82.45 ± 2.48** | 59.65 ± 8.94 | 75.43 ± 6.56 | |
| Shape | 1 | **32.00 ± 4.00** | 20.00 ± 0.00 | 26.00 ± 12.00 | **93.33 ± 4.71** | 60.00 ± 7.49 | 33.20 ± 4.71 | |
| | 5 | **88.00 ± 9.80** | 60.00 ± 0.00 | 48.00 ± 7.50 | **96.66 ± 4.71** | 85.67 ± 2.45 | 85.39 ± 12.47 | 100.00 |
| | 10 | **84.00 ± 8.00** | 62.00 ± 7.48 | 48.00 ± 4.00 | **100.00 ± 0.00** | 88.67 ± 4.61 | 87.36 ± 4.71 | |
| BA-Motif | 1 | **73.00 ± 7.38** | 61.2 ± 8.08 | 67.60 ± 4.52 | **71.66 ± 2.49** | 50.63 ± 0.42 | 67.60 ± 4.52 | |
| | 5 | **89.00 ± 1.67** | 83.4 ±10.67 | 49.60 ± 1.96 | **96.00 ± 1.63** | 82.54 ± 0.87 | 77.60 ± 2.21 | 100.00 |
| | 10 | **91.60 ± 3.72** | 79.01 ± 1.34 | 50.60 ± 1.56 | **98.00 ± 0.00** | 90.89 ± 0.22 | 84.33 ± 2.49 | |
| BA-LRP | 1 | **64.72 ± 4.44** | 49.52 ± 0.43 | 51.12 ± 2.50 | 71.56 ± 3.62 | 54.11 ± 5.33 | **77.48 ± 1.21** | |
| | 5 | **85.50 ± 2.05** | 79.01 ± 1.35 | 49.87 ± 1.28 | **91.60 ± 1.57** | 59.21 ± 0.99 | 77.76 ± 0.52 | 97.95 |
| | 10 | **95.50 ± 0.50** | 56.97 ± 1.10 | 52.38 ± 1.79 | **94.90 ± 1.09** | 66.40 ± 1.47 | 88.38 ± 1.40 | |
| Graph-Twitter | 10 | **58.13 ± 2.74** | 49.47 ± 0.96 | 46.59 ± 5.85 | **56.43 ± 1.39** | 40.00 ± 3.98 | 52.40 ± 0.29 | |
| | 50 | **59.73 ± 1.11** | 55.67 ± 1.04 | 50.20 ± 5.71 | **58.93 ± 1.29** | 55.62 ± 1.12 | 52.92 ± 0.27 | 61.40 |
| | 100 | 53.25 ±1.30 | **59.76 ± 1.00** | 56.65 ± 2.78 | **59.51 ± 0.31** | 53.37 ± 0.55 | 55.47 ± 0.51 | |
| Graph-SST5 | 10 | **36.62 ± 0.76** | 28.06 ± 0.33 | 29.33 ± 3.25 | **35.72 ± 0.65** | 25.49 ± 0.39 | 24.90 ± 0.60 | |
| | 50 | 37.64 ± 0.83 | 35.96 ± 1.04 | **37.83 ± 3.62** | **43.81 ± 0.86** | 31.47 ± 2.58 | 23.15 ± 0.35 | 44.39 |
| | 100 | **42.05 ± 1.35** | 41.04 ± 0.79 | 41.87 ± 1.80 | **44.43 ± 0.45** | 32.01 ± 1.90 | 25.26 ± 0.75 | |

**Table 3: *Predictive Accuracy* when generating 10 interpretive graphs per class.**

| Dataset | Graph-Twitter | Graph-SST5 | BA-Motif | BA-LRP | Shape | MUTAG | XGNN on MUTAG |
|---|---|---|---|---|---|---|---|
| GNNInterpreter | 74.40 ± 0.06 | 88.60 ± 0.09 | 100.00 ± 0.00 | 85.00 ± 0.00 | 100.00 ± 0.00 | 70.00 ± 0.00 | 100.00± 0.00 |
| GDM | 91.11 ± 0.02 | 91.33 ± 0.00 | 100.00 ± 0.00 | 95.50 ± 0.00 | 100.00 ± 0.00 | 86.67 ± 0.047 | |

**Table 4: Ablation study showing *Model Fidelity* when generating 10 interpretive graphs per class.**

| Dataset | Graph-Twitter | Graph-SST5 | BA-Motif |
|---|---|---|---|
| GDM-First | 25.84±4.06 | 21.28±0.21 | 51.20±2.23 |
| GDM-Last | 28.61±3.41 | 27.19±0.27 | 46.40±2.53 |
| GDM-Ensemble | 30.68±6.00 | 25.70±0.25 | 51.40±5.56 |
| **GDM** | **58.13±2.74** | **36.62 ±0.76** | **91.60±3.72** |
| Dataset | BA-LRP | Shape | MUTAG |
| GDM-First | 51.03±0.75 | 60.00±0.00 | 73.68±2.19 |
| GDM-Last | 49.95±0.28 | 60.00±1.00 | 56.84±5.78 |
| GDM-Ensemble | 56.39±0.54 | 58.00±0.00 | 87.37±0.73 |
| **GDM** | **95.50±0.50** | **64.00 ±8.00** | **94.73±0.00** |

model snapshot would capture less feature and structure information, thus the model fidelity score would be smaller than GDM as shown in Table 4. In the ablation study, there are also notable discrepancies between the GDM-Ensemble fidelity and GDM fidelity on a few datasets, including Graph-Twitter, BA-Motif, and BA-LRP. Those ensemble snapshots would possibly preserve misleading patterns which could be filtered out during model training but been captured while distribution matching, leading to the large deviates of the fidelity score for the GDM-Ensemble model. Generally, we can observe that the distribution matching design is effective: disabling this design will greatly deteriorate the performance.

**Parameter Sensitivity** In our final objective Eq. (9), we defined two hyper-parameters $\alpha$ and $\beta$ to control the strength for feature matching and sparsity matching regularization, respectively.

In this subsection, we explore the sensitivity of hyper-parameters $\alpha$ and $\beta$. Since MUTAG is the only dataset that contains node features, we only apply the feature matching regularization on this dataset. we vary the sparsity coefficient $\beta$, and report the utility and predictive accuracy for all of our datasets in Figure 3. For most datasets excluding Shape, the utility performance start to degrade when the $\beta$ becomes larger than 0.5. This means that when the interpretive graph becomes more sparse, it will lose some information during training time. Given the small values of $\beta$, the graphs are relatively dense and the model predictive accuracy for all datasets except Graph-SST5 and Graph-Twitter converges to be stationary, denoting that the sparsity of those graphs would not heavily influence generating interpretations.

Moreover, we report the model utility and model fidelity with different feature-matching coefficients $\alpha$ in Table 5. A larger $\alpha$ means we have a stronger restriction on the node feature distribution. We found that when we have more strict restrictions, the utility increases slightly. This is an expected behavior since the node features from the original MUTAG graphs contain rich information for classifications, and matching the feature distribution enables the interpretation to capture rare node types. By having such restrictions, we successfully keep the important feature information in our interpretive graphs. However, as the coefficient $\alpha$ increase, the model fidelity would slightly decrease, which means the restrictions about

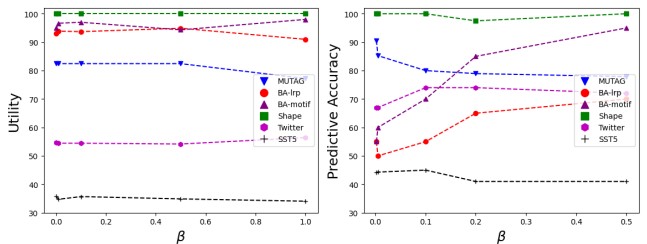

**Figure 3: Sensitivity analysis of hyper-parameter $\beta$ for the sparsity matching regularization**

**Table 5: Sensitivity analysis of hyper-parameter $\alpha$ for feature matching regularization.**

| $\alpha$ | 0.005 | 0.01 | 0.05 | 0.5 | 1.0 |
|---|---|---|---|---|---|
| Model Utility | 82.45 | 82.45 | 82.45 | 82.45 | 80.70 |
| Model Fidelity | 78.94 | 78.95 | 65.31 | 63.16 | 68.42 |

feature distribution would impact the model embeddings and sparsity and the ideal interpretive graphs are generated by balancing these restrictions.

### 4.4 Qualitative Analysis

We qualitatively visualize the global interpretations provided by GDM to verify that GDM can capture human-intelligible patterns, which indeed correspond to the ground-truth rules for discriminating classes. Table 6 shows examples in BA-Motif, MUTAG, BA-LRP and Shape datasets, and more results and analyses on other datasets can be found in Appendix A.4.

The qualitative results show that the global explanations successfully identify the discriminative patterns for each class. If we look at BA-Motif dataset, for the house-shape class, the interpretation has captured such a pattern of house structure, regardless of the complicated base BA graph in the other part of graphs; while in the non-house class with five-node cycle, the interpretation also successfully grasped it from the whole BA-Motif graph. Regarding the Shape dataset, the global interpretations for all the classes are almost consistent with the ground-truth graph patterns, i.e., Wheel, Lollipop, Grid and Star shapes are also reflected in the interpretation graphs. Note that the difference for interpretative graphs of Star and Wheel are small, which provides a potential explanation for our fidelity results in Table 3, where pre-trained GNN models cannot always distinguish interpretative graphs of Wheel shape with interpretative graphs of Star shape.

### 5 CONCLUSIONS

In this work, we studied a new problem to enhance interpretability for graph learning: how to provide global interpretation for a model training procedure, such that training on such interpretation can recover a similar model? We proposed a novel framework, where interpretations are optimized based on the whole training trajectory. We designed an interpretation method GDM via distribution

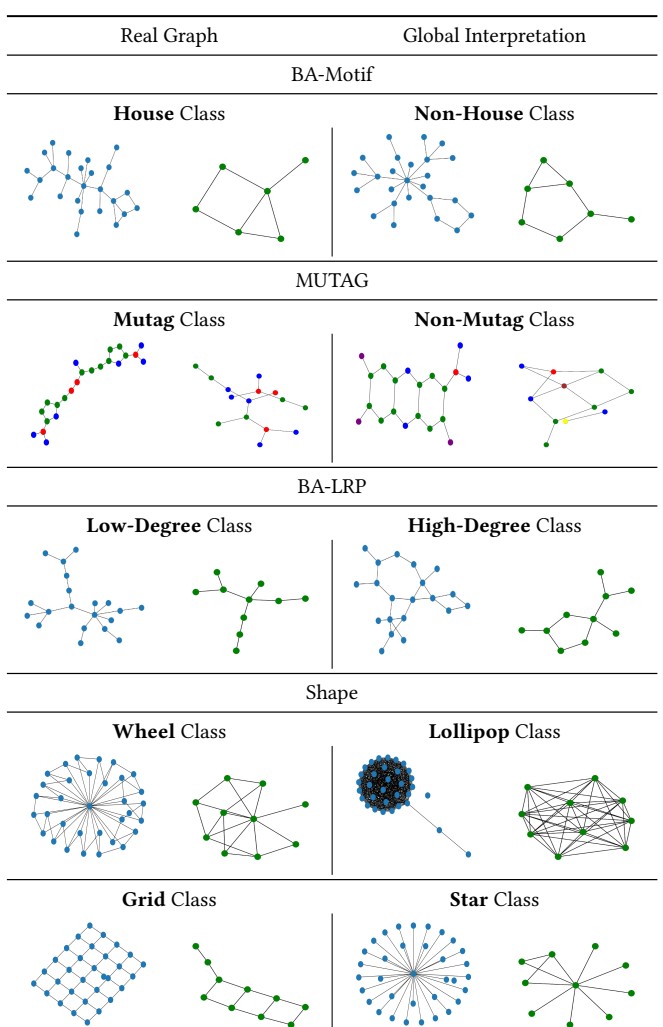

**Table 6: Visualization of real example graphs and generated global interpretations**

matching, which matches the embeddings generated by the GNN model for the training data and interpretive data. Our framework can generate an arbitrary number of interpretable and effective interpretive graphs, and could be easily integrated in the model training pipline. We evaluated our method both quantitatively and qualitatively on real-world and interpretive datasets. Besides existing metrics, we proposed new metric *model fidelity* to evaluate the fidelity of the model trained on interpretive graphs. The results indicate that the explanation graphs produced by GDM achieve high performance and are able to capture class-relevant structures by probing the training trajectory and demonstrate efficiency. One possible limitation of our work is that the interpretations are a general summarizing of the whole training procedure, thus cannot reflect the dynamic change of patterns captured by the model to help detect anomalous behavior, which we believe is an important and challenging open problem. In the future work, we aim to extend the proposed framework to study model training dynamics.

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

# A  APPENDIX

## A.1  GDM versus Inherently Interpretable Model

We compare the performance of GDM with a simple yet inherently global-interpretable method, logistic regression with handcrafted graph-based features. When performing LR on these graph-structured data, we leverage the Laplacian matrix as graph features: we first sort row/column of adjacency matrix by nodes' degree to align the feature dimensions across different graphs; we then flatten the reordered laplacian matrix as input for LR model. When generating interpretations, we first train a LR on training graphs and obtain interpretations as the top most important features (i.e. edges on graph) based on regression weights where average number of edges. We then report the utility of LR interpretations, shown in the following table 7.

### Table 7: Model Utility of Logistic Regression

| Dataset | MUTAG | BA-Motif | BA-LRP | Shape | Graph-Twitter | Graph-SST5 |
|---|---|---|---|---|---|---|
| LR Interpretation | 93.33% | 100% | 100% | 100% | 42.10% | 22.68% |
| Original LR | 96.66% | 100% | 100% | 100% | 52.06% | 27.45% |

LR shows good interpretation utility on simple datasets like BA-Motif and BA-LRP, but it has much worse performance on more sophisticated datasets, compared with GDM in table **??**. For example, interpretations generated by GDM can achieve close accuracy as the original GCN model.

## A.2  GDM versus Local Interpretation

As aforementioned, GDM provides global interpretation, which is significantly different from the extensively studied local interpretation methods: we only generate few small interpretive graphs per class to reflect the high-level discriminative patterns the model captures; while local interpretation is for each graph instance. Though our global interpretation is not directly comparable with existing local interpretation, we still compare their interpretation utility to demonstrate the efficacy of our GDM when we only generate a few interpretive graphs. The results can be found in Table 8. We compare our model with GNNExplainer[1],PGExplainer[2] and Captum[3] on utility. For Graph-SST5 and Graph-Twitter, we generate 100 graphs for each class and 10 graphs for other datasets.

### Table 8: Utility Compared with Local Interpretation

| Datasets | Graph-SST5 | Graph-Twitter | MUTAG | BA-Motif | Shape | BA-LRP |
|---|---|---|---|---|---|---|
| GNNExplainer | 43.00±0.07 | 58.12±1.48 | 73.68±5.31 | 93.2±0.89 | 89.00±4.89 | 58.65±4.78 |
| PGExplainer | 28.41 ± 0.00 | 55.46 ± 0.03 | 75.62±4.68 | 62.58±0.66 | 71.75±1.85 | 50.25±0.15 |
| Captum | 28.83±0.05 | 55.76±0.42 | **89.20±0.01** | 52.00±0.60 | 80.00±0.01 | 49.25±0.01 |

Comparing these results, we can observe that the GDM obtains higher utility score compared to different GNN explaination methods, with relatively small variance.

---

[1]https://github.com/RexYing/gnn-model-explainer
[2]https://github.com/flyingdoog/PGExplainer
[3]https://github.com/pytorch/captum

### Table 9: *Efficiency* when generating 10 interpretive graphs per class.

| Dataset | Graph-Twitter | Graph-SST5 | BA-Motif | BA-LRP |
|---|---|---|---|---|
| Time (s) | 169.29 | 291.36 | 184.89 | 155.41 |

| Dataset | Shape | MUTAG | XGNN on MUTAG | |
|---|---|---|---|---|
| Time (s) | 176.01 | 218.45 | 838.20 | |

## A.3  Time Efficiency

The Table 9 shows the time consumed for generating 10 interpretive graphs per class on all datasets by GDM, and the time needed for generating graphs on MUTAG by XGNN

## A.4  More Qualitative Results

**MUTAG** This dataset has two classes: "non-mutagenic" and "mutagenic". As discussed in previous works [4, 28], Carbon rings along with $NO_2$ chemical groups are known to be mutagenic. And [16] observe that Carbon rings exist in both mutagen and non-mutagenic graphs, thus are not really discriminative. Our synthesized interpretive graphs are also consistent with these "ground-truth" chemical rules. For 'mutagenic" class, we observe two $NO_2$ chemical groups within one interpretative graph, and one $NO_2$ chemical group and one carbon ring, or multiple carbon rings from a interpretative graph. For the class of "non-mutagenic", we observe that $NO_2$ groups exist much less frequently but other atoms, such as Chlorine, Bromine, and Fluorine, appear more frequently. To visualize the structure more clearly, we limit the max number of nodes to 15 such that we do not have too complicate interpretative graphs.

**BA-Motif and BA-LRP** The qualitative results on BA-Motif dataset show that the explanations for all classes successfully identify the discriminative features. For House-shape class, all the generated graphs have captured such a pattern of house structure, regardless of the complicated base BA graph in the other part of graphs. For the other class with five-node cycle, our generated graphs successfully grasp it from the whole BA-Motif graph. In BA-LRP dataset, the first class consists of Barabasi-Albert graphs of growth parameter 1, which means new nodes attached preferably to low degree nodes, while the second class has new nodes attached to Barabasi-Albert graphs preferably to higher degree nodes. Our interpretative dataset again correctly identify discriminative patterns to differentiate these two classes, which are the tree-shape and ring-shape structures.

**Shape** In Table 10, the generated explanation graphs for all the classes are almost consistent with the desired graph patterns. Note that the difference for interpretative graphs of Star and Wheel are small. This provides a potential explanation for our post-hoc quantitative results in Table 2, where pre-trained GNN models cannot always distinguish interpretative graphs of Wheel shape with interpretative graphs of Star shape.

Received 12 October 2023; revised 20 December 2023; accepted 29 December 2023

| Dataset | Class | Training Graph Example | Synthesized Interpretation Graph |
|---|---|---|---|
| **BA-Motif** | House |  |  |
| | Non-House |  |  |
| **BA-LRP** | Low Degree |  |  |
| | High Degree |  |  |
| **MUTAG** | Mutagenicity |  |  |
| | Non-Mutagenicity |  |  |
| **Shape** | Wheel |  |  |
| | Lollipop |  |  |
| | Grid |  |  |
| | Star |  |  |

**Table 10: The qualitative results for all datasets. For each class in each dataset, as a reference, the example graph selected from the training data is displayed in the left column, while the generated explanation graphs are in the right column. Different node colors in MUTAG represent different node types.**

