# OpenReview forum: "Globally Interpretable Graph Learning via Distribution Matching"
_ACM.org/TheWebConf/2024/Conference — TheWebConf24_

### Official Review · Reviewer_3rF4 · 2023-11-05

**Novelty:** 6
**Technical Quality:** 5

**Review:**

This paper studied a novel globally interpretable graph learning problem. The goal of globally interpretable graph learning was to provide global interpretation such that training on such interpretation could recover a similar model. To solve this problem, this paper introduced a Graph Distribution Matching (GDM) approach which aligned the feature distributions between the original graphs and the interpretation graphs.

**Quality:** This paper first analyzed the limitations of existing global interpretation methods in terms of model fidelity. It was found in Figure 1 that existing global interpretation methods suffered in preserving the predictive logits though achieving satisfactory prediction accuracy. This motivated the Graph Distribution Matching (GDM) approach by considering the model training trajectory.

**Clarity:** The motivation and the proposed GDM approach were clearly explained. The empirical results on real-world data sets also supported the effectiveness of GDM in terms of both model fidelity and prediction accuracy.

**Originality:** A novel globally interpretable graph learning problem was formulated by enforcing the similarity between the predictive logits of the original model and that of the surrogate model. By simulating the model training trajectory, the proposed GDM could preserve the model fidelity and prediction accuracy when generating the interpretation graphs.

**Significance:** As stated in this paper, the proposed GDM could benefit both the model developer and the consumers in practice. Developers would publish transparent models without leaking raw data, and consumers would have a better understanding of the prediction behavior on their own data sets.

The strengths of this paper are given below.

(S1) The model fidelity metric is introduced to evaluate the trustworthiness of the interpretation graphs.

(S2) A novel Graph Distribution Matching (GDM) approach is presented by synthesizing interpretive graphs from a distribution matching perspective. This allows to preserve the intermediate information of GNNs in the feature space.

(S3) Experimental results demonstrate the effectiveness of the GDM over baselines, and the qualitative analysis also confirms that the generated interpretive graphs are positively correlated with human-intelligible patterns.

The weaknesses of this paper are given below.

(W1) The optimization problem in Eq. (2) is confusing. The training trajectory w.r.t. $\Phi_t$ ($t=1,2,\cdots$) is determined by raw input graphs $G$ and gradient descent optimizer. In this case, $\Phi_t$ can be pre-computed with a fixed order. It is unclear regarding the difference between GDM and GDM-Ensemble (disabling the inner optimization). What are the differences of optimization problems between GDM and GDM-Ensemble in generating interpretation graphs?

(W2) The connection between the distribution matching loss in Eq. (4) and MMD can be explained. As shown in line 365, MMD is defined in the Hilbert kernel space. But Eq. (4) involves the difference in the feature space learned by $f_{\theta_t}$. Does it indicate that the feature space learned by $f_{\theta_t}$ is a Hilbert kernel space?

(W3) Table 5 shows that the feature matching regularization might not be necessary for GDM, because GDM achieved good performance when $\alpha$ approaches 0. More results are required to validate the necessity of the feature-matching regularization in real-world graphs.

**Questions:**

(Q1) It is shown in Table 3 that XGNN has a promising performance on MUTAG. It might be convincing to report the performance of XGNN on MUTAG in Table 2 for model comparison.

(Q2) Is it feasible to derive theoretical connections between Graph Distribution Matching in Eq. (4) and fidelity/accuracy preservation? For example, why can simulating the feature distribution trajectory of GNNs improve the quality of synthesized graphs in terms of model fidelity and predictive accuracy?

**Reviewer Confidence:**

3: The reviewer is confident but not certain that the evaluation is correct

**Scope:**

4: The work is relevant to the Web and to the track, and is of broad interest to the community

---

### Official Review · Reviewer_hhBu · 2023-11-22

**Novelty:** 5
**Technical Quality:** 5

**Review:**

This paper proposes a new interpretation metric model fidelity for evaluating the fidelity of the resulting model trained on interpretations. Based on the metric, the paper proposes a graph distribution matching framework to find the most informative pattern the model captures during training.

**Questions:**

Strength:
1. The idea of the paper is reasonable and interesting. How to learn a global interpretation especially patterns captured by the model for a certain class is really needed for the graph tasks.
2. The paper proposes different interpretation metrics to evaluate the model, such as model fidelity, model utility and predictive accuracy.
3. The paper evaluated the model on both synthetic and real-world datasets for graph classification. The better performance demonstrates the effectiveness of the proposed method.

Weakness:
1. The parameter of the method is large especially when the average edge number of the graph is high. It is better to add the space cost analysis of the model.
2. How to get the graph patterns ground truth of a class in section 4.4, it is better to add the details in experimental settings.
3. It is better to visualize the different snapshot of the learned graph structure to demonstrate the effectiveness of the model in ablation study.

**Reviewer Confidence:**

3: The reviewer is confident but not certain that the evaluation is correct

**Scope:**

4: The work is relevant to the Web and to the track, and is of broad interest to the community

---

### Official Review · Reviewer_61Kc · 2023-11-24

**Novelty:** 4
**Technical Quality:** 4

**Review:**

Strengths of the Paper:
1. Global interpretation is an excellent entry point; even as research on GNN interpretability has progressed, there are very few studies on global interpretation. I believe global interpretation is a highly significant topic.
2. The extensive experiments validate the effectiveness of the proposed method, and I am pleased to see the authors conduct many qualitative experiments. In my view, qualitative experiments are of greater importance than quantitative experiments in the field of interpretability.
3. The idea based on distribution matching is innovative, and I believe it can provide valuable references and inspiration for subsequent development of global interpretation algorithms.

Weaknesses of the Paper:
1. My major concern is the large number of parameters to be optimized, as the features and adjacency matrix of the entire interpretable subgraph S are trainable, which will undoubtedly bring tremendous computational and storage pressure. Therefore, during the reading process, I was hoping to find some tricks for accelerated or approximate calculation, but none have been found so far. I think the authors need to consider this issue, as the task of this paper is different from graph compression. Graph compression only needs to generate a compressed subgraph for graph data, while global interpretation requires generating a model interpretation subgraph for each category and each graph neural network, which undoubtedly greatly intensifies the demand for computing power.
2. My second concern is that the proposed interpretation method is different from traditional methods such as SA, GradCAM, GNNExplainer, and PGExplainer. Specifically, these traditional methods allocate an importance score to each node or edge in the graph, and then synthesize the interpretable subgraph according to the requirements of the task (i.e., the sparsity level of the interpretation needed). This is reasonable because no additional prior needs to be introduced during the process of assigning importance scores. However, does the proposed method need to pre-define the size of S? This poses a risk: if the size of S set is smaller than the actual size of the ground truth subgraph, will the effect still be good? At the same time, I believe this situation is highly likely to occur, for example, in graphs like MNIST-SP with a large number of nodes.
3. In Section 3.5 of the paper, I think the design of Matching Edge Sparsity and Matching Feature Distribution requires more explanation. Since this paper aims to find global interpretations during the model training process, I think it is reasonable for the sparsity and model features to differ significantly from the original data, even when the training trajectory is completely matched. Therefore, I think further clarification of the motivation for adding these two modules is needed here.
4. The global interpretation algorithm GLGExplainer [1] is mentioned in the method section, and I think it should also be compared as a baseline in the experiments. This would contribute to the completeness of the experiments in the paper. [1] Global Explainability of GNNs via Logic Combination of Learned Concepts. ICLR23

**Questions:**

See the weaknesses.

**Reviewer Confidence:**

3: The reviewer is confident but not certain that the evaluation is correct

**Scope:**

4: The work is relevant to the Web and to the track, and is of broad interest to the community

---

### Official Review · Reviewer_eVZw · 2023-11-24

**Novelty:** 5
**Technical Quality:** 4

**Review:**

In this paper, the authors propose a novel framework called Graph Distribution Matching (GDM) to provide global interpretations of graph neural network (GNN) models throughout the training process. The paper formulates globally interpretable graph learning as optimizing interpretations to match the distribution of graphs along the entire training trajectory of the GNN, not just the final trained model. Evaluation on classification benchmarks demonstrates that GDM achieves both high model fidelity in recovering predictive behaviors and accuracy in classifying interpretations, outperforming prior global interpretation methods. Qualitative analysis also shows interpretable patterns discovered by GDM.

Pros:

+ Introduces a new problem formulation of globally interpretable graph learning that explains the model training procedure, rather than just the final model. This is a meaningful and interesting perspective.
+ Achieves both high model fidelity and predictive accuracy in interpretations, outperforming previous global interpretation baselines. The efficiency is also notable.

Cons:

- It would be beneficial if the related work could cover some dataset condensation literature.
- Quantitative evaluation only considers predictive performance metrics; evaluation of interpretation quality, i.e., how interpretable the generated graphs are, is lacking.
- Some technical details are missing. See questions below.

**Questions:**

1. How do the authors prevent the mode collapse problem?
2. The concept of "fidelity" has been used in many explanation works. What is the key difference between the proposed "model fidelity" and previous definitions?
3. It seems that Algorithm 1 corresponds to a sequential optimization problem, rather than Equation 2: at step ( t ), previous snapshots are discarded and will no longer contribute to the final result. In this case, should we be concerned about the catastrophic forgetting problem?
4. The authors have discussed the performance of GDM-First and GDM-Ensemble in Table 4. Why does GDM-Last perform poorly? As only one snapshot was used, did GDM and GDM-Last have the same number of training iterations?

**Reviewer Confidence:**

3: The reviewer is confident but not certain that the evaluation is correct

**Scope:**

3: The work is somewhat relevant to the Web and to the track, and is of narrow interest to a sub-community

---

### Official Review · Reviewer_zEC4 · 2023-11-24

**Novelty:** 6
**Technical Quality:** 4

**Review:**

Strengths
- Motivation is great; Interpretation from the perspective of model developers/providers is novel. Tracking the training curves also broadens the scope of interpretation.
- The authors proposed to focus on how discriminative the generated interpretive graphs are in terms of recovering the original model's training procedure, which I believe is a good point for interpretation.
- Novel loss that keeps tracking the distribution gaps during training in addition to the end of training.

Weakness/questions
- Vague definition: the model fidelity, model utility, and predictive accuracy were described vaguely by texts. Especially, all these three metrics are defined by the authors (predictive accuracy seems to choose different ones from XGNN and GNNInterpreter; if not, please correct me) and do not provide formal definitions, it weakens the credibility of the results.
- Definition of predictive accuracy: As far as I understand, the authors measure the original model's accuracy on the interpretive graphs in terms of the original labels, not the predicted labels (or probabilities) on the original graphs. Then, if we produce a simple graph that any GNN can easily achieve high accuracy, we will always have the perfect predictive accuracy.  Could you explain this potential problem?
- In Figure 1, why is the model fidelity of GDM also decreasing as training proceeds? If this metric is supposed to decrease as the training proceeds, I'm not sure this is the right metric to measure the interpretation. In contrast, predictive accuracy increases as the interpretive graphs become similar to the original one, as we expected.

**Questions:**

Described above.

**Ethics Review Description:**

Nothing

**Reviewer Confidence:**

3: The reviewer is confident but not certain that the evaluation is correct

**Scope:**

4: The work is relevant to the Web and to the track, and is of broad interest to the community

---

### Decision · Program_Chairs · 2024-01-22

**Decision:**

Accept

**Comment:**

This paper explores the domain of global interpretation for graph neural networks (GNNs) during the training process, introducing a novel framework called Graph Distribution Matching (GDM). The strengths lie in the paper's motivation, unique perspective on model development, and tracking training curves. The proposed focus on discriminative interpretive graphs and the introduction of a novel loss add depth to the interpretation process. However, weaknesses include vague definitions of model fidelity, utility, and predictive accuracy, potentially compromising result credibility. Additionally, concerns are raised about the computational and storage demands due to the large number of parameters, and the need for more explanation in certain sections, such as Matching Edge Sparsity and Matching Feature Distribution.
 Overall, the paper has valuable contributions but requires improvements in clarity, definition precision, and addressing the raised concerns.